# Digital finance and M&As: An empirical study and mechanism analysis

**Ziyu Jiang**[1,2], **Xihao Sun**[3], **Yan Song**[4], **Guojian Ma**[1] *

**1** Management School, Jiangsu University, Zhenjiang, Jiangsu, China, **2** Program on Chinese Cities, University of North Carolina at Chapel Hill, Chapel Hill, North Carolina, United States of America, **3** Basic Education Department, Taihu University of Wuxi, Wuxi, Jiangsu, China, **4** Department of City and Regional Planning, University of North Carolina at Chapel Hill, Chapel Hill, North Carolina, United States of America

* 1000001788@ujs.edu.cn

**Data Availability Statement:** The data underlying the results presented in the study are available from China Stock Market & Accounting Research Database (https://www.gtarsc.com/), WIND database (https://www.wind.com.cn/portal/en/

## Abstract

With the rapid growth and wide application of digital technology, enterprises have entered the digital era with both opportunities and challenges existing. Mergers and acquisitions are one of the most efficient ways to integrate resources and achieve profit growth, giving enterprises advantages in competing in the new mode of economic growth. Based on this, this research tries to explore whether the development of digital finance will contribute to the emergence of M&As activities through combining M&As data of the Chinese stock market with the digital finance inclusion index between 2012 and 2020. The results show that the development of digital finance largely influences M&As activities through lower acquirers' financial constraints. We further replace digital finance with three sub-indexes including coverage breadth, usage depth, and digitalization level to explore the impact of different dimensions of digital finance on M&As. Results show that coverage breadth plays a more important role. In addition, heterogeneity tests reveal that the relationship between the development of digital finance and M&As activities varies significantly. The influences of digital finance on private and western and central enterprises are more significant compared with state-owned and eastern enterprises. According to the study, since the development of digital finance can be an efficient way to ease financial constraints and boost M&As activities, the government should promote the development of digital finance while companies strive to make the most use of it.

## 1. Introduction

Currently, the global economy is experiencing a slowing down period or even entering the recession. Apart from that, the influence caused by the COVID-19 pandemic has not faded and the Russia-Ukraine conflict is still going on. Plenty of enterprises struggle in an economy ravaged by these factors with profits being squeezed and it is getting harder to survive. Therefore, enterprises urgently seek ways to achieve transformation and grow innovatively. Mergers and acquisitions (M&As) have been proved efficient in helping enterprises to obtain resources, to expand their market share, and to make more profits [1–3]. According to the statistics

WDS/database.html)and Institute of Digital Finance
Peking University (https://idf.pku.edu.cn/yjcg/zsbg/
513800.htm)

**Funding:** Unfunded studies

**Competing interests:** NO authors have competing
interests

released by the IMAA [4], the number of M&As in China reached 8,000 in 2016, with a transaction value of about USD 1,000 billion in 2015 since 2000 (see Fig 1). Many companies have taken M&As as a core strategy to pursue rapid growth and become more competitive.

With the development and proliferation of digital technology, it is possible for enterprises to develop and operate digital businesses, and those who with new digital business models would be more compatible in the context of the digital economy. Based on the statistics and investigation of IDC, the majority of enterprise organizations (at 53%) have an enterprisewide digital transformation strategy [5]. They also forecast 2023 will become the inflection point of enterprise digital transformation, and the next five years will be a booming period for digital development.

After digitalization has eventually come to be taken seriously and the Chinese government now views digitalization as part of its national policy, thereby promoting the modernization of its economic system. The Chinese government published "The 14th Five-Year Plan for Digital Economy Development" on 12 January, 2022 [6] and placed an emphasis on exploiting new competitive advantages through digitalization. Digital finance as a new form of financial format with the core of digital transformation grows rapidly. Through combining big data, artificial intelligence, and cloud computing, digital finance allows enterprises to solve financing problems and gain sufficient funds, thereby giving them advantages on M&As and achieving both the desired transformation in order to develop further.

Recent research has already proved that the development of digital finance can boost economic growth, producing large-scale economic benefits with respect to innovation, entrepreneurship, investment, and social security including employment rate, income levels, and consumption. [7] found that the development of inclusive digital finance had had a significant role in promoting rural entrepreneurship. [8–10] also studied the relationship between digital finance and entrepreneurship and reached the same conclusions. While [11] discovered that digital finance had had a positive effect on urban innovation. [12] also find that digital finance can boost SME innovation by easing financing constraints, which is also proved by [13–16]. Besides, [17] reached the conclusion that the development of digital finance can contribute to household consumption mainly on recurring household expenditures and less developed regions. [18] focused on how to better regulate the income distribution of residents through digital finance and make Meanwhile, which is also consistent with [19]. Elsewhere, [20] stated that digital finance had had a significant positive impact on the servitization of the manufacturing industry in China.

It is obvious that the focus of digital finance has mainly been on regional development, households, and employment until now. However, there has been a lack of attention paid to its influence on enterprise development including M&As. Will digital development bring benefits with regard to M&As? What mechanism enables digital finance to place an effect on M&As? These questions need further research.

This study tries to examine the relationship between digital finance and M&As, looking specifically to clarify the effects of digitalization, and contributing to M&A research in several aspects. First, this paper analyzes whether the development of digital finance is likely to have an influence on M&A activities. Second, we try to examine the mechanism enabling digital finance to affect M&As with a particular emphasis on the relevant financial constraints. In addition, we also try to investigate the extent of the heterogeneity in different digital financial structures, regions, equity nature, and sectors, thereby extending the research on digital finance to a more micro level, seeking to provide reliable advice for enterprises seeking to achieve digital transformation.

The structure of this paper proceeds as follows: Section 2 outlines relevant theories and presents a review of the literature related to digital finance, M&As, and financial constraints,

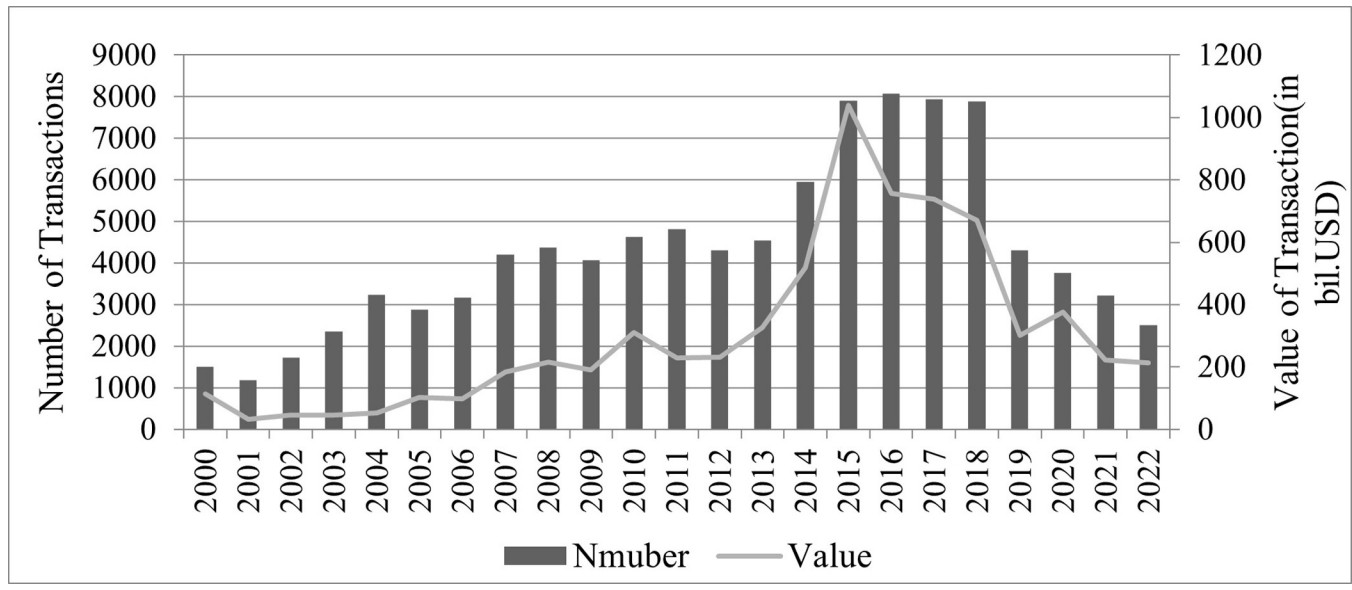

**Fig 1. Mergers and acquisitions in China.**

before developing the hypotheses; Section 3 describes the data and methodology; Section 4 presents and discusses the empirical results as well as the robustness test; Section 5 contains further discussion about the mechanisms through which digital finance influences M&As; and, finally, Section 6 concludes the research and proposes possible recommendations for the government and managers.

## 2. Theory and hypotheses

In this section, we discuss the relevant theory, previous literature and accordingly develop hypotheses with regard to the relationship between M&As, digital finance, and financial constraints.

### 2.1 Digital finance and M&As

In the classic research case of a used car market, Akerlof found that buyers were at a disadvantage due to information asymmetry. [21] also reached the same conclusion from their research on the insurance market. Pertinently, the asymmetric distribution of information to parties or the incompleteness of information for one side can prevent M&A activities and reduce the efficiency of market operations. Information asymmetry may even lead to the failure of some potentially valuable M&As and unrealized profits, meaning that shareholders are not often inclined to favor M&A strategies, as already proved by [22, 23]. Meanwhile, [24] suggested that, in fact, serious information asymmetry exists in every element of M&A transactions.

The first aspect is when acquirers do not know the real situation within target companies, sometimes leading to an overestimation of the transaction value. The industry and regional heterogeneity of the acquirer and the target may have a bearing on the level of information asymmetry here [25]. Compared with horizontal M&As, vertical M&As tend to experience greater information asymmetry due to the heterogeneity of the market, the production technology, and the differences between upstream and downstream enterprises. Similarly, M&As in certain regions experience more serious information asymmetry as the level of development

and marketization varies from one region to the next. In addition, regional protectionism leads to poor information flow, which exacerbates the problem.

The second aspect regarding information asymmetry concerns employee turnover and cultural differences between the involved companies. Acquirers may have to undergo an outflow of core staff and can find it hard to maintain managerial or operational efficiency. Besides, as most M&As are one-off deals, in order to obtain greater compensation, the target companies are motivated to hide key information and to even release false information.

[26] held that in the social economic system, the more distrust between transacting parties, the more guarantees needed in the transaction, leading to higher costs in general and a higher premium being paid by the acquirer. However, digital finance can alleviate information asymmetry to a large extent and make transactions more transparent. Normally, financial platforms not only accumulate large-scale financial data, but they also offer advantages in data modeling and application from the collection of both structured and unstructured data. Digital finance can allow for an analysis of the sellers' operational data comprehensively through big data technology and intensive business processes, thus putting together an accurate picture of the sellers' risks. As a result, managers tend to make M&As decisions only after acquiring sufficient and accurate information.

## 2.2 Digital finance, financial constraints, and M&As

Research until now has held that the digital finance increases M&A performance largely through alleviating financial constraints. When enterprises try to expand the scale of their business or enter a new market, large costs need to be paid. Clearly, those lacking a sufficient cash flow need to access finance from elsewhere. The ability to acquire enough money is a key factor determining the success of M&As. Indeed, researchers believe that financial constraints will decrease the likelihood of M&A activities [27]. Thus, financial capacity has a significant influence on M&As, which is consistent with [28, 29]. However, under the traditional financing model, the financing needs of most enterprises cannot be met due to a credit mismatch [30]. To be more specific, there are two main types of mismatches: supply and demand mismatch and domain mismatch. Normally, small and medium-sized companies and those engaged in traditional industries struggle to acquire funds due to a lack of collateral. According to the long tail theory, there are plenty of long-tail customers in financial markets. Thus, the problem of low financing efficiency of traditional financial modes seriously restricted the transformation of economic structure and high-quality development of enterprises. The emergence of digital finance has largely eased the difficulty and cost of financing though. In particular, through combining big data, artificial intelligence, as well as blockchain, and other information technologies, digital finance has been alleviating the problems of a high-risk premium and high operating cost caused by information asymmetry in the traditional financing model [31]. Digital finance also provides stable technical support to expand the scale and accessibility of financial services [32]. Digital payment, peer-to-peer lending and other forms of business have largely reduced searching and risk identification costs, largely eliminating the intermediary role of financial institutions. As a result, enterprises' financing costs can be reduced, and the financing period is shortened too.

In terms of effects and mechanisms, digital finance mainly eases financial constraints in two regards: incremental supplements and stock optimization. The former means that digital finance effectively absorbs financial resources in the market and provides them precisely for enterprises with actual needs. [33] stated that digital finance can process a massive amount of data at a low cost. Therefore, financing costs are largely lower compared with the traditional financing model. Meanwhile, stock optimization refers to the in-depth optimization of the

quality and efficiency of traditional financial institutions and businesses. [34] pointed out that peer-to-peer lending subverts the traditional credit pricing model through the increased transparency and informatization of credit. As a kind of financial spillover, digital finance helps to break through financing boundaries and avoids a credit structure mismatch.

With the above analysis in mind, we propose the following hypotheses:

H1: The development of digital finance can encourage a greater number of M&As activities.

H2: Digital finance can increase the emergence of M&As activities by easing enterprises' financial constraints.

## 3. Research design and methodology

### 3.1 Sample selection and data source

This study focuses on domestic M&As in mainland China between 2012–2020, with companies listed in the SH and SZ stock exchanges chosen. M&A data are collected from the WIND database while financial data have been obtained from the China Stock Market & Accounting Research (CSMAR) database. Digital finance data have been acquired from the Digital Financial Inclusion Index compiled by the Digital Finance Research Center of Peking University [35, 36]. The samples also needed to meet the following requirements: (1) ST and ST* listed firms are excluded; (2) financial and insurance sectors are not considered; (3) the transaction value has to be larger than ¥ 100 million and the ratio of deal size to the acquirer's total assets must be at least 1% but not more than 150%; (4) failed M&A transactions are excluded; and (5) samples with insufficient data are excluded. Furthermore, if a company announced multiple M&A transactions within the year, its largest deal is selected. This paper also winsorized variables in the 1st and 99th percentiles to lower the effect of extreme values. Overall, 10,883 samples were selected.

### 3.2 Variables

The dependent variable is M&As, which is a dummy variable. The score is 0 when companies have not conducted M&A activities, and 1 is given after when it has conducted at least one M&A. The independent variable is the development of digital finance *(DF)*, which is measured according to the Digital Finance Inclusion Index. The Digital Finance Research Center of Peking University compiled this index to measure the development of digital finance at the province and city levels, excluding Hong Kong, Macau, and Taiwan [35, 36]. Based on the consumption data provided by the Ant Financial Group, 33 indicators are selected with a traditional inclusive finance index compilation method to construct a digital finance index addressing coverage breadth, usage depth, and level of digitalization. Learning from recent research, we have taken the Digital Financial Inclusion Index as the proxy variable for the development of digital finance.

In terms of financial constraints, this research also takes it as mediators. At present, there are some disputes ongoing about the measurement of financing constraints between scholars. Instead of traditional accounting ratios, indicators including KZ index [37], WW index [38], and SA index [39] are more widely accepted for measuring financial constraints. According to [40], the KZ index and the WW index use endogenous variables when constructing, which may cause deviation. Thus, this research has applied the SA index to measure companies'

financial constraints. The relevant formula can be expressed as follows:

$$SA = -0.737*Size + 0.043*Size^2 - 0.04*Age \tag{1}$$

Where Size is the natural logarithm of the company's total assets, while Age is the difference between the observation year and the registration year of the company. Therefore, the outcome of the SA index is less than 0, the larger the score is, the greater the financing constraints will be.

In order to minimize the deviation caused by missing variables, drawing on previous research, this research has included the following control variables at the corporation level: (1) Acquiring proportion *(Proportion)*. The shareholding ratio that acquired from the owner. (2) Acquirers' market value *(MV)*. Since the size of companies is relatively large, it is measured by its natural log. (3) Company's age *(Age)*. Measured by the difference between the current year and the year of establishment. (4) Leverage *(LEV)*. This is represented by the ratio of total debt divided by total assets. (5) The ability to generate profits *(EBIT)*. This is calculated as the natural log of earnings before interest and taxes. (6) Ownership concentration (*Ownership*). This is the shareholding ratio of the top ten shareholders within the company. (7) Whether the chairman and CEO of the acquirer is the same person *(Separation)*. If the two positions are held by the same person, a score of 0 is given, while otherwise, it is 1. A summary of variables is shown below in Table 1.

## 3.3 Model specification

Before running the linear regression test, we first adopt VIF correlation test to examine whether there are multicollinearity issues. In order to examine whether the development of digital finance will contribute to M&A activities and given that the dependent variable is a dummy variable, this research has selected the binary Probit model as the base model to test H1, and the regression model (2) is set below. To examine H2, this research uses the bootstrap method to examine the mediating effect, and the regression model is presented in Eqs (2) to (4). If 0 is not contained in the confidence interval, then it can be concluded that the mediating effect of financial constraints exists.

$$Probit(MA_{it} = 1) = \alpha_1 + \alpha_2 DF_{it} + \sum \alpha_3 \text{ Controls}_{it} + \sum year + \sum ind + \varepsilon_1 \tag{2}$$

$$SA = \beta_1 + \beta_2 DF_{it} + \sum \beta_3 \text{ Controls}_{it} + \sum year + \sum ind + \varepsilon_2 \tag{3}$$

$$Probit(MA_{it} = 1) = \theta_1 + \theta_2 DF_{it} + \theta_3 SA + \sum \theta_4 \text{ Controls}_{it} + \sum year + \sum ind + \varepsilon_3 \tag{4}$$

In the above equations, subscripts i and t represent industry and time respectively. The dependent variable MA is. The independent variable DF represents the level of digital finance development in the company's province in year t. While controls refer to the control variables, and $\varepsilon$ is the random error. In addition, this paper also fixes the effects of industry and time effects to control for possible interference in the regression results of unobserved factors at different levels. The coefficient $\alpha_2, \beta_2, \theta_2$ of the independent variable $DF_{it}$ reflects the overall impact of the development of digital finance on M&As activities, which is expected to be significantly positive according to the hypothesis above.

**Table 1. Summary of variables.**

| Variable type | Variable name | Symbol | Definition |
|---|---|---|---|
| Dependent variable | M&As | MA | Pre MA = 0, MA = 1 |
| Independent variable | Digital finance | DF | The development of local digital finance |
| Mediator | Financial constraints | SA | SA = -0.737×Size+0.043×Size2-0.04×Age |
| Control variable | Acquiring proportion | Proportion | The shareholding ratio acquired. |
| | Market value | Market value | Measured by the natural log of market value. |
| | Age | Age | The difference between the current year and the year of establishment. |
| | Leverage | Lev | The ratio of total debt divided by total assets. |
| | Profitability | EBIT | The natural log of earnings before interest and tax. |
| | Ownership concentration | Ownership | Shareholding ratio of top 10 shareholders. |
| | Whether the chairman and CEO of acquirers are the same person | Separation | If two positions are held by the same person, then 0 is given, otherwise 1 is set. |

# 4. Empirical analysis

## 4.1 Descriptive statistical features

The descriptive statistical results of the main variables in this research are shown in Table 2. It can be found that digitization grows rapidly between 2012 and 2020. The mean value of digital finance is 265.8168, with the maximum value being 431.9276, the minimum value being 18.84, and the standard deviation being 67.6133. There are significant differences in financial constraints among listed companies. The maximum value of financial constraints is -0.6839 while the minimum value is -5.1025, and the mean value is -3.7550. the statistical results of the other control variables, such as leverage and profitability, are close to the existing research results and are within a reasonable range.

## 4.2 Correlation analysis

Generally speaking, there may be a multicollinearity issue when the vif between explanatory variables is higher than 10. The results in Table 3 indicate that all the value of vif is below 10, which can prove that there is no multicollinearity issue existing in this study and the regression model would not be affected.

**Table 2. Summary of descriptive features.**

| Variables | Min | Max | Mean | Std. |
|---|---|---|---|---|
| DF | 18.84 | 431.9276 | 265.8168 | 67.6133 |
| SA | -5.1025 | -0.6839 | -3.7550 | 0.2692 |
| Proportion | 1 | 100 | 56.5613 | 34.8810 |
| Market value | 8.6399 | 13.3656 | 10.0004 | 0.5403 |
| Age | 4 | 49 | 17.7833 | 5.7756 |
| Lev | -1.6242 | 6270.051 | 3.2576 | 76.4856 |
| EBIT | 3.8942 | 11.5602 | 8.4541 | 0.6573 |
| Ownership | 8.26 | 100 | 59.8881 | 14.8458 |
| Separation | 0 | 1 | 0.3112 | 0.4630 |

**Table 3. VIF correlation test.**

| Variable | VIF | 1/VIF |
|---|---|---|
| MV | 4.33 | 0.231029 |
| EBIT | 3.99 | 0.250522 |
| Age | 3.23 | 0.309354 |
| SA | 3.14 | 0.318314 |
| Ownership | 1.12 | 0.891274 |
| Separation | 1.04 | 0.962070 |
| Proportion | 1.01 | 0.991414 |
| Lev | 1.00 | 0.997976 |
| Mean VIF | 2.36 | |

## 4.3 Regression analysis

The influence of the development of digital finance on merger and acquisition activities is shown in Table 4 below. Model (1) only considers the relationship between digital finance and M&As without adding other control variables while Model (2) is the result with control variables. It can be found that the coefficient of DF is 0.0044 and 0.0050 respectively and significantly positive at 1% level, indicating that the development of digital finance can largely contribute to the emergence of M&As activities. Therefore, hypothesis 1 is true.

**Table 4. Regression results of digital finance.**

| Variable | Model (1) | Model (2) |
|---|---|---|
| DF | 0.0044*** | 0.0050*** |
| | (33.63) | (32.55) |
| Proportion | | 0.0017*** |
| | | (6.20) |
| Market value | | 0.4508*** |
| | | (12.43) |
| Age | | -0.200*** |
| | | (-11.45) |
| Lev | | 0.0001 |
| | | (0.52) |
| EBIT | | -0.2102*** |
| | | (-7.22) |
| Ownership | | -0.0044*** |
| | | (-6.36) |
| Separation | | -0.0370 |
| | | (-1.77) |
| Industry | Yes | Yes |
| Time | Yes | Yes |
| Constant | -1.1807*** | -3.4928*** |
| | (-32.65) | (-16.90) |
| Obs. | 10883 | 10883 |
| $R^2$ | 0.0389 | 0.0513 |

Note: (1) ***, **, and * represent significant at 1%, 5%, and 10% levels respectively. (2) The values in parentheses are z values.

## 4.4 Mediating analysis

The results of the mediating analysis are represented in Table 5 below. It can be found that the coefficients of DF of model (2) and (3) are 0.0050 and -0.002 separately, both significant at 1% level. After adding financial constraints in the model (4), the result still reaches the 1% significant level with the coefficients of DF being 0.0078. In addition, according to bootstrap examination results, the confidence interval of the indirect effect is between 0.0005 and 0.007, and 0 is not included, which indicates that financial constraints play a mediating effect in how the development of digital finance contributes to M&As activities. Thus, H2 has been verified.

## 4.5 Robustness test

In order to ensure the robustness of results, we run various robustness tests. First, we change the estimation model. Since the dependent variable M&As is a dummy variable, we select the Logit model to do the robustness test and results are shown in model (5) of Table 6 below. It is not hard to find that the coefficient of DF is 0.0081, positive and significant at 1% level, which shows the development of digital finance still has significant driving effects on M&As activities. Therefore, research conclusions do not change according to the model setting. Second, M&As activities are largely associated with the macroeconomic environment. Due to the spread of Coivd-19 worldwide, we currently are experiencing a period of economic stagnation and a larger number of economic activities have to be suspended. Thus, in order to ensure the robustness of results, this research minimizes the number of samples. We delete the sample in 2020 and the time period is limited to 2012 to 2019. Results in model (6) indicate that the significance is unchanged, thus hypothesis 1 is still positive. Third, we use SA index to measure financial constraints when analyzing the mechanism of how digital finance influences M&As. In the robustness test, we select KZ index to replace SA index to examine whether hypothesis 2 is still true. The results in model (7) show that the coefficient of DF is 0.0050 and still significant at 1% level, which proves that there is no variation in the conclusion of "Digital finance can increase the M&As activities by lowering enterprises' financial constraints".

Besides, endogeneity is also a key problem that may exist in this study. This research may miss potential variables that affect both dependent and independent variables, which may lead to the endogenous problem of "reverse causality" or "omitted variables". On the one hand, M&As activities will put an influence on the development of digital finance. On the other hand, except control variables, there may be other important variables which may affect

**Table 5. Regression results of mediating test.**

| Variable | (2) MA | (3) SA | (4) MA |
|---|---|---|---|
| DF | 0.0050*** (32.55) | -0.0002*** (-11.5008) | 0.0078*** (30.4889) |
| SA | | | -3.0337*** (-22.1174) |
| Control | Yes | Yes | Yes |
| Industry | Yes | Yes | Yes |
| Time | Yes | Yes | Yes |
| Constant | -3.4928*** (-16.90) | -4.6923*** (-196.6487) | -19.7879*** (-27.5613) |
| Obs. | 10883 | 10883 | 10883 |
| Indirect effect | | [0.0005, 0.0007] | |

Note: (1) ***, **, and * represent significant at 1%, 5%, and 10% levels respectively. (2) The values in parentheses are t/z values.

**Table 6. Results of robustness test.**

| Variable | Model (5) | Model (6) | Model (7) | Model (8) | Model (9) |
|---|---|---|---|---|---|
| DF | 0.0081*** (31.81) | 0.0059*** (34.34) | 0.0050*** (32.40) | 0.0052*** (33.57) | 0.0046*** (20.84) |
| Control | Yes | Yes | Yes | Yes | Yes |
| Industry | Yes | Yes | Yes | Yes | Yes |
| Time | Yes | Yes | Yes | Yes | Yes |
| KZ | | | 0.0205*** (4.04) | | |
| Interest rate | | | | | -0.0599*** (-2.75) |
| _Cons | -5.6678*** (-16.74) | -3.6621*** (-16.76) | -3.3526*** (15.17) | -3.5699*** (-17.23) | -3.2135*** (-13.96) |
| Obs. | 10883 | 10291 | 10883 | 10883 | 10883 |
| $R^2$ | 0.0512 | 0.0556 | 0.0522 | 0.0544 | 0.0516 |

Note: (1) ***, **, and * represent significant at 1%, 5%, and 10% levels respectively. (2) The values in parentheses are z values.

M&As and digital finance at the same time. For example, macroeconomic factors such as policy, and economic situation can affect the development of digital finance and M&As decisions, resulting in deviation in the estimation of regression coefficients. Therefore, we select the average development level of digital finance in the year as the instrumental variable to test whether endogenous problems exist in this research. The average development level of digital finance can influence digital finance index based on the city level, but it has no direct impact on M&As decisions. It can be found in Model (8) that the coefficient of DF is 0.0052, still significant at 1% level and consistent with former results, which proves the result is robust. In addition, this paper also examines the impact of missing macroeconomic factors. We add risk-free interest rate in the regression model as it can reflect the macroeconomic environment when M&As occur. Figures in model (9) show that the significance of DF does not change after adding macroeconomic factors, thus the conclusion is convincing.

## 5. Further discussion

Considering the heterogeneity of digital financial structure, the imbalance of regional economic development, and differences in enterprises' equity nature and industries, these may lead to great differences in the impact of digital finance on M&As activities in different enterprises. Therefore, this research divides digital finance into three dimensions and classifies the samples into several categories according to regions, equity nature, and sector differences to do further analysis.

### 5.1 The heterogeneity of digital financial structure

The digital finance index covers three sub-dimensions including coverage breadth, usage depth, and digitalization level. Coverage breadth reflects the regional financial environment by counting the number of people using electronic accounts. Usage depth describes the regional financial services' ability through the usage of various digital financial services while digitalization level measures the cost and efficiency of digital finance. In order to more accurately examine which dimension plays a more significant role on enterprises' M&As activities, we carry out structural analysis of digital finance. The results are shown in Table 7.

It can be found that in Table 7, the significances of all these three dimensions reach the 1% significant level. Further, we conduct the seemingly unrelated estimation test (SUEST) to

**Table 7. Results of the heterogeneity of digital financial structure.**

| Variable | Model (10) | Model (11) | Model (12) |
|---|---|---|---|
| coverage | 0.0223*** (60.84) | | |
| usage | | 0.0116*** (46.34) | |
| digitalization | | | 0.0171*** (57.80) |
| Industry | Yes | Yes | Yes |
| Time | Yes | yes | Yes |
| Control | Yes | Yes | Yes |
| Constant | -6.3738*** (-16.20) | -6.8040*** (-18.03) | -5.3799*** (-13.74) |
| Obs. | 10883 | 10883 | 10883 |
| $R^2$ | 0.1969 | 0.1102 | 0.1766 |

Note: (1) ***, **, and * represent significant at 1%, 5%, and 10% levels respectively. (2) The values in parentheses are Z values.

compare whether differences in coefficients exist. As SUEST can only compare whether the coefficients differed between two groups [41], we perform pairwise comparisons between the coefficients of coverage, usage, and digitalization. Results in Table 8 indicate that the p values are all significant at 1% level while the coefficients of these three dimensions are 0.0223, 0.0116, and 0.0171 respectively. Therefore, it can be concluded that coverage breadth plays a much stronger role, followed by digitalization level. The breadth of digital financial coverage is more conducive to improving broaden the boundaries of financial services and from the perspective of efficiency, coverage breadth is the most direct way to benefit enterprises in remote areas. Therefore, more digital infrastructure should be constructed at the early stage, through expanding the coverage of digital services to strengthen usage depth and promote digitalization level. [42] also proved that the larger the coverage breadth is, the more customers can be covered, which is also proved by [43].

## 5.2 Region, digital finance and M&As

The unbalanced distribution of financial resources broadens the economic gap between different regions, which leads to the east more developed while the central and west being less developing. Besides, according to "China regional financial operation report (2021)" [44] issued by The People's Bank of China, the number of banking financial institutions in western and middle China was 59,000 and 53,000 respectively, compared with more than 100,000 in the east.

**Table 8. Results of SUEST test.**

| Variable | Usage | Digitalization | East | SOE | Second | Third |
|---|---|---|---|---|---|---|
| Coverage | 0.000*** | 0.000*** | | | | |
| Usage | | 0.000*** | | | | |
| Non-east | | | 0.000*** | | | |
| Private | | | | 0.001*** | | |
| First | | | | | 0.1193 | 0.2220 |
| Second | | | | | | 0.1287 |

Note: (1) ***, **, and * represent significant at 1%, 5%, and 10% levels respectively.

As a result, the financial demand of most enterprises especially in west and central areas cannot be met.

According to the criteria for the division of east, west, and central areas in China issued by the National Bureau of Statistics, we divide samples into two categories based on the regions where companies registered. 0 is given to companies located in eastern region, while central and western areas are set as 1. Based on the analysis above, this paper proposes Hypothesis 3: The influences of digital finance on M&As activities are more significant for enterprises located in west and central areas.

As Column (1)—(2) shown in Table 9, it can be found that the significances of both east and non-east areas are at 1% significant level. SUEST test indicates that the p value is 0.000 and significant at 1% level, which proves the influences various on different areas. Since the coefficients of east and Midwest areas are 0.0379 and 0.1008 separately, it can be concluded that the effects of the development of digital finance in promoting M&As activities are much stronger in the west and central area. Thus, H3 is approved. After receiving the same level of financial supports, Midwest enterprises are more likely to implement M&As activities. Therefore, the promotion of the development of digital finance is important for narrowing the regional financial gap and giving Midwest companies more opportunities for achieving sustainable development and industrial transformation. The findings are also consistent with [20, 45–49].

## 5.3 Equity nature, digital finance, and M&As

[50] suggest that the financial system in China is dominated by the banking system which is controlled by the government indeed. Although state-owned enterprises may suffer from financing problems to some extent, due to financial subsidies and policy supports they received from the governments, state-owned enterprises are much easier to get financing. Besides, private companies are mainly small and medium-sized compared with state-owned enterprises, which means they have fewer assets and collateral. Furthermore, private enterprises have lower management efficiency and quality of financial information. Thus, it is difficult for private enterprises to get funds from traditional financial markets. While the development of digital finance can attract more investors and provide more targeted financial products and services at the same time, the financial demands of private enterprises can be met to a larger extent. Therefore, this paper believes private enterprises are more likely to conduct M&As activities if financial constraints can be alleviated. Similarly, the value of 0 is given to state-owned enterprises, otherwise, the variable is set as 1. Accordingly, we propose Hypothesis 4: Compared with state-owned enterprises, the positive effects of the development of digital finance on M&As activities are more prominent for private enterprises.

Column (3) and (4) in Table 9 presents the results of the impact of digital finance on M&As activities across companies with different equity nature. It shows that the significance of both state-owned and private enterprises is at the 1% level and the p value of SUEST test is 0.001, which is positive and significant at 1% level. The coefficients of state-owned and private enterprises are 0.0253 and 0.0287 individually, therefore, the development of digital finance put greater influences on private enterprises as their financial demands are much larger, thus supporting H4. Private enterprises make up an important role of China's economy and the digital finance enhance their access to financial services. As a result, private enterprises operate and produce more efficiently, which leads to a more dynamic economy. The result is also consistent with [51–54].

**Table 9. Regional heterogeneity test results.**

| Variable | Region | | Equity nature | | Sector | | |
|---|---|---|---|---|---|---|---|
| | (1) | (2) | (3) | (4) | (5) | (6) | (7) |
| | East | Non-east | SOE | Private | First | Second | Third |
| DF | 0.0379*** | 0.1008*** | 0.0253*** | 0.0287*** | 0.0431*** | 0.0268*** | 0.0281*** |
| | (62.23) | (38.21) | (34.05) | (55.87) | (6.38) | (52.99) | (37.53) |
| Control | Yes | Yes | Yes | Yes | Yes | Yes | Yes |
| Industry | Yes | Yes | Yes | Yes | Yes | Yes | Yes |
| Time | Yes | Yes | Yes | Yes | Yes | Yes | Yes |
| Constant | -8.3418*** | -9.7294*** | -4.7314*** | -7.3064*** | -1.7503 | -9.3662*** | -5.1855*** |
| | (-15.36) | (-6.50) | (-6.07) | (-12.85) | (-0.29) | (-8.60) | (-5.76) |
| Obs. | 8098 | 2785 | 3217 | 7666 | 101 | 7101 | 3681 |
| $R^2$ | 0.3105 | 0.5964 | 0.2120 | 0.2494 | 0.3032 | 0.2322 | 0.2436 |

Note: (1) ***, **, and * represent significant at 1%, 5%, and 10% levels respectively. (2) The values in parentheses are Z values.

## 5.4 Sector, digital finance, and M&As

According to "Guidelines for the classification of listed companies in China" (2012 Version) [55], we divide samples into three categories based on industry attributes and do regression tests. Specifically, companies in category A are classified into primary sector mainly agricultural enterprises in China. Industrial manufacturing industry enterprises belonging to category B are considered as secondary sector. The remaining enterprises are regarded as third sector. Currently, agricultural enterprises are at the mature stage and have lower capital demands, providing funds for banks and other financial institutions instead. On the contrary, industrial companies need plenty of funds to operation to achieve sustainable development and industrial transformation. Also, most of the non-financial tertiary industries are private enterprises and companies' sizes are relatively small, which makes them difficult to get funds from traditional financial institutions. Besides, primary industry has fewer M&As activities compared with secondary and third industries due to their operation strategies. Thus, we propose Hypothesis 5: The positive effects of the development of digital finance on M&As activities are less prominent on primary industry compared with secondary and third industry.

The regression results are presented in Column (5), (6), and (7) of Table 9. It can be found that the significances of primary, secondary, and tertiary industry are all at 1% level with the coefficients of 0.0431, 0.0268, and 0.0281 respectively. However, SUEST test in Table 8 shows that none of the p values are significant with the figures of 0.1193, 0.2220, and 0.1287 individually, which means there is no difference in the impact of digital finance on different sectors. Thus, Hypothesis 5 should be not accepted. One possible explanation is that, according to statistics, the number of M&As in the primary industry is relatively small compared with the other two industries. After setting several sample selection criterions, the eligible sample size is only 101, which is not large enough to find the difference across sectors and may lead to variations in results. While a recent research [56] indicates that the impact of the development of digital finance on M&As activities varies across industries. To be more specific, the positive effects of the development of digital finance on M&As activities are more prominent in the tertiary industry with the primary coming to the second. However, their findings still differ from our hypothesis. Thus, it points out the way for future analysis.

## 6. Conclusion and suggestion

### 6.1 Conclusion

Digitalization has already become a vital strategy for boosting economic growth, while the development of digital finance provides a new opportunity for the emergence of M&As activities. This research takes M&As events in China mainland between 2012 and 2020 as samples to explore whether the development of digital finance will encourage M&As activities. Through empirical examination, we reach the following conclusions. First, the development of digital finance can put a significant positive influence on M&As activities, which largely increases the possibility of conducting M&As events. Second, the mechanism of how the development of digital finance contributes to M&As activities is through lowering financing constraints. After considering the model setting and estimation bias, the robustness of core variables, the sample validity, and endogeneity, the effects of the development of digital finance on M&As activities do not change significantly. Apart from that, heterogeneity tests show that coverage breadth plays a much stronger role than usage depth and digitalization level. Differences in equity nature and locations lead to large differences in the emergence of M&As activities, and such differences affect the role of digital finance in driving M&As activities similarly. Our research indicates that the impact of the development of digital finance on state-owned and eastern enterprises' M&As activities has clear boundary limits. On the contrary, private, western, and central enterprises are more likely to implement M&As activities. Besides, although the coefficients of the first, second, and third industry vary, SUEST test shows that there are no differences between the impact of the development of digital finance on M&As activities across sectors.

### 6.2 Suggestion

The former research conclusions have certain reference values for improving M&As success rate and performance and achieving industry upgrading and transformation. Thus, we propose some practical implications for both the Chinese government and enterprises.

First, the government should pay more attention to the advantages of backwardness brought by the development of digital finance and accelerate innovation in digital financial derivatives. Currently, a new round of technological revolution represented by digitalization is advancing, which spawns a series of new financial formats. The development of innovative financial products such as online lending platforms and e-commerce supply chains effectively reduces financing constraints and provides a wide range of support for enterprises.

Second, since the significance of coverage breadth is much greater, the government should strengthen the IT infrastructure and promote the development of business flexibility [57]. However, as the results shown that the significance of both usage depth and digitalization reach the 1% significant level, the development of these two dimensions should not be ignored as well. The digital finance cannot provide solid support for enterprises if the development of digital finance only focuses on coverage breadth. In terms of regional differences, due to the inadequate financial infrastructure, low distribution density of financial institutions, and Internet penetration in Midwest areas, the development of digital finance is hindered in these areas. Therefore, the government should improve the construction of digital infrastructure and narrow the "digital gap". Through combining the advantages of low-cost and rich resources of Midwest area and the advantages in technology, market, and digital industries of the east area, a digital financial development system that complements and coordinates with each other can be formed. As a result, digital finance can better benefit enterprises in remote areas.

Considering the relationship between cost and efficiency, traditional financial modes only focus on leading companies while the financial demands of private and SMEs are not resolved. Hence, the government should issue preferential policies for private and SMEs, especially those who have accumulated certain core technologies and innovative potentials, guiding traditional financial institutions to offer additional credit facilities to long-tail customers. At the same time, the government can encourage the traditional banking sector to innovate in financial services by combining digital technologies and rebuilding the financing system guided by fintech. As a result, the credit demand of enterprises can be met to a larger extent.

What is more, the government and regulatory authorities should take systemic risks of digital finance into consideration and formulate regulatory policies to prevent the occurrence of systemic risks. As the combination of finance, technology, and networks, the rapid development of digital finance improves the efficiency of financial resource allocation, bringing great convenience for private and SMEs. However, it also weakens the boundary between regions, industries, and financial institutions, which may lead to systematic risks. Thus, the government and regulatory authorities are supposed to set up criterions to assess risks and formulate macro-policy to control them accordingly.

For enterprises, they should grasp the opportunities brought by the development of the digital economy and make full use of them to acquire diversified financing channels. Also, SMEs are supposed to improve the reliability of information and transfer efficiency. Through improving internal governance and developing incentive measures, managers are encouraged to disclose real and reliable information. Hence, the problem of information asymmetry can be reduced, accompanied by the quality of internal control improved, which will largely lower the financing difficulties.

## Supporting information

**S1 File. Dataset.**
(XLSX)

## Author Contributions

**Conceptualization:** Ziyu Jiang, Xihao Sun.

**Methodology:** Ziyu Jiang.

**Writing – original draft:** Ziyu Jiang.

**Writing – review & editing:** Yan Song, Guojian Ma.

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
