## [Decision Letter · Decision Letter 0]

24 Apr 2023

PONE-D-23-09479Digital finance and M&As: An empirical study and mechanism analysisPLOS ONE

Dear Dr. ma,

Thank you for submitting your manuscript to PLOS ONE. After careful consideration, we feel that it has merit but does not fully meet PLOS ONE’s publication criteria as it currently stands. Therefore, we invite you to submit a revised version of the manuscript that addresses the points raised during the review process.

We look forward to receiving your revised manuscript.

Kind regards,

Stefan Cristian Gherghina, PhD. Habil.

Academic Editor

PLOS ONE

Journal Requirements:

   "Unfunded studies"

4. Please ensure that you refer to Figure 1 in your text as, if accepted, production will need this reference to link the reader to the figure.

5. We note you have included a table to which you do not refer in the text of your manuscript. Please ensure that you refer to Table 1 and 6 in your text; if accepted, production will need this reference to link the reader to the Table.

Additional Editor Comments:

The article needs to be revised several times considering the theoretical and empirical approach.

Reviewers' comments:

Reviewer's Responses to Questions

**Comments to the Author**

1. Is the manuscript technically sound, and do the data support the conclusions?

Reviewer #1: Yes

Reviewer #2: Yes

Reviewer #3: Yes

2. Has the statistical analysis been performed appropriately and rigorously? 

Reviewer #1: Yes

Reviewer #2: No

Reviewer #3: Yes

3. Have the authors made all data underlying the findings in their manuscript fully available?

Reviewer #1: Yes

Reviewer #2: No

Reviewer #3: Yes

4. Is the manuscript presented in an intelligible fashion and written in standard English?

Reviewer #1: Yes

Reviewer #2: Yes

Reviewer #3: Yes

5. Review Comments to the Author

Reviewer #1: The digital finance affecting M&As is a good topic by the data of 2012-2020 from DFRCoPKU, SH and SZ stock exchanges. The process of analysis, discussion and heterogeneity tests is normal, the results are positive mostly, while the abstract is simple without concrete data background, questions, conclusion, and heterogeneity analysis results.

Reviewer #2: This manuscript uses data from Chinese listed companies, and the Digital Financial Inclusion Index compiled by the Digital Finance Research Center of Peking University (2020), validating that the development of digital finance largely contributes to M&As activities through lower acquirers’ financial constraints. The manuscript focuses on the popular topic of digital economy development, which has very important practical significance. The research method is standardized, the logic is clear, the structure is reasonable and complete, and the conclusion has strong persuasiveness and reference value.

The deficiencies of this article or the areas that need to be improved are mainly as follows:

First, although the results of various regressions are significant, they only report three decimal places, almost all of which are 0.000. The specific results cannot be seen and need to be corrected. The units can be modified, or more decimal places can be reported.

Second, in the robustness test, the (5) regression, after replacing the dummy variable with the Probit model, the regression result is still very small. Then, the development of local digital finance has a significant impact on M&A behavior, but to what extent? This requires further proof.

Third, in the heterogeneity analysis, the three aspects of the digital financial structure are all significant, how can we see which is more important? Or are all three aspects important? Enterprises should invest in more efficient fields in the early stage of investment, whether there is a difference in time series.

Fourth, when analyzing the heterogeneity of Equity nature, is there a large gap between the coefficients of state-owned and private companies, and can it be concluded that the development of local digital finance has a greater impact on private companies? These issues require further argument.

Fifth, In the analysis of industry-level heterogeneity, the coefficient of the primary industry is larger. Why is it said that the impact on the secondary and tertiary industries is greater than that of the primary industry.

Reviewer #3: Dear authors, Thank you for giving me the opportunity to read your interesting research. The paper is of really good quality, talking about the overall contribution and statistics. However, there are several points that should be improved:

1. The abstract of the paper is missing information about the method used.

2. In-text citations are absolutely incorrect, they did not follow any standard. You can either use author+year or the number in the brackets - please revise throughout the paper. 

3. Introduction - this section of the paper provides all necessary information about the previous papers, main aim and contribution (please, correct only the in-text citations).

4. Theory and hypotheses development - I do appreciate this section of the paper, which uses current references to develop the research hypotheses. 

5. Research and methods - please, clarify why 2012-2020 was chosen. Why is 2020 included? Is not this year significantly affected by the COVID-19 pandemic?

It would be appropriate to present some information about the dataset (basic descriptive characteristics).

Section 3.2 does not need to be too structured. 

Equations 3, 4, 5 - please explain all variables.

The methodological steps are missing, there is no information about the methods used in this section. Please add. 

6. Were the preconditions of the regression analysis  verified? Please give details. 

7. The results are explained very succinctly - please check and revise.

8. Please, focus more on the discussion section, use more references to discuss your findings in the context of other relevant studies published online.

6. PLOS authors have the option to publish the peer review history of their article (what does this mean?). If published, this will include your full peer review and any attached files.

Reviewer #1: No

Reviewer #2: No

Reviewer #3: No

---

## [Author Response · Author response to Decision Letter 0]

8 Jun 2023

Rebuttal letter

Dear editor:

Thank you so much for your advice on the manuscript entitled “Digital finance and M&As: An empirical study and mechanism analysis” (Manuscript ID: PONE-D-23-09479). Your comments are very valuable and helpful for revising the paper and improving the quality. Details of revision are provided below:

First, the manuscript has been thoroughly copyedited by alba editing & proofreading. Second, “The authors received no specific funding for this work.” Besides, regarding to point 4 and 5, changes have been made according to the requirements.

We did our best to revise the paper and please read the whole revised manuscript. Thank you again for your valuable advice concerning our manuscript.

Dear reviewer 1:

Thank you so much for your comments on the manuscript entitled “Digital finance and M&As: An empirical study and mechanism analysis” (Manuscript ID: PONE-D-23-09479). Your comments are very valuable and helpful for revising the paper and improving the quality. Details of revision are provided below.

The process of analysis, discussion and heterogeneity tests is normal, the results are positive mostly, while the abstract is simple without concrete data background, questions, conclusion, and heterogeneity analysis results.

Response: Thanks for the comment. We have extended the abstract. The revised part is presented below:

With the rapid growth and wide application of digital technology, enterprises have entered in the digital era with both opportunities and challenges existing. Mergers and acquisitions are one of the most efficient ways to integrate resources and achieve profit growth, gives enterprises advantages in competing in the new mode of economic growth. Based on this, this research tries to explore whether the development of digital finance will contribute to the emergence of M&As activities through combining M&As data of Chinese stock market with digital finance inclusion index between 2012 and 2020. The results show that the development of digital finance largely influence M&As activities through lower acquirers’ financial constraints. We further replace digital finance with three sub-indexes including coverage breadth, usage depth and digitalization level to explore the impact of different dimensions of digital finance on M&As. Results show that coverage breadth plays a more important role. In addition, heterogeneity tests reveal that the relationship between the development of digital finance and M&As activities varies significantly. The influence of digital finance on private and western and central enterprises are more significant compared with state-owned and eastern enterprises. According to the study, since the development of digital finance can be an efficient way to ease financial constraints and boost M&As activities, the government should promote the development of digital finance while companies strive to make the most use of it.

Dear reviewer 2:

Thank you so much for your comments on the manuscript entitled “Digital finance and M&As: An empirical study and mechanism analysis” (Manuscript ID: PONE-D-23-09479). Your comments are very valuable and helpful for revising the paper and improving the quality. Details of revision are provided below.

1. First, although the results of various regressions are significant, they only report three decimal places, almost all of which are 0.000. The specific results cannot be seen and need to be corrected. The units can be modified, or more decimal places can be reported.

Response: Thanks for the comment. We have made some changes as followed:

The influence of the development of digital finance on merger and acquisition activities are shown in Table 4 below. Model (1) only considers the relationship between digital finance and M&As without adding other control variables while Model (2) is the result with control variables. It can be found that the coefficient of DF is 0.0044 and 0.0050 respectively and significantly positive at 1% level, indicating that the development of digital finance can largely contribute to the emergence of M&As activities. Therefore, hypothesis 1 is true.

Table 4 Regression results of digital finance

Variable Model (1) Model (2)

DF 0.0044***

(33.63) 0.0050***

(32.55)

Proportion 

 0.0017***

(6.20)

Market value 0.4508***

(12.43)

Age -0.200***

(-11.45)

Lev 0.0001

(0.52)

EBIT -0.2102***

(-7.22)

Ownership -0.0044***

(-6.36)

Separation -0.0370

(-1.77)

Industry Yes Yes

Time Yes Yes

Constant -1.1807***

(-32.65) -3.4928***

(-16.90)

Obs. 10883 10883

R2 0.0389 0.0513

Note: (1) ***, ** and * represent significant at 1%, 5% and 10% levels respectively. (2) The values in parentheses are z value.

2. Second, in the robustness test, the (5) regression, after replacing the dummy variable with the Probit model, the regression result is still very small. Then, the development of local digital finance has a significant impact on M&A behavior, but to what extent? This requires further proof.

Response: Thanks for the comment. We have added the significance as followed:

In order to ensure the robustness of results, we run various robustness tests. First, we change the estimation model. Since the dependent variable M&As is a dummy variable, we select Logit model to do robustness test and results are shown in model (5) of Table 6 below. It is not hard to find that the coefficient of DF is 0.0081, positive and significant at 1% level, and the development of digital finance still has a significant driving effect on M&As activities. Therefore, research conclusions do not change according to the model setting.

3. Third, in the heterogeneity analysis, the three aspects of the digital financial structure are all significant, how can we see which is more important? Or are all three aspects important? Enterprises should invest in more efficient fields in the early stage of investment, whether there is a difference in time series.

Response: Thanks for the comment. The heterogeneity of digital financial structure has been fully revised, please have a look:

Digital finance index covers three sub-dimensions including coverage breadth, usage depth and digitalization level. Coverage breadth reflects the regional financial environment by counting the number of people using electronic accounts. Usage depth describes the regional financial services ability through the usage of various digital financial services while digitalization level measures the cost and efficiency of digital finance. In order to more accurately examine which dimension plays a more significant role on enterprises’ M&As activities, we carry out structural analysis of digital finance. The results are shown in Table 7.

Table 7 Results of the heterogeneity of digital financial structure

Variable Model (10) Model (11) Model (12)

coverage 0.0223***

(60.84) 

usage 0.0116***

(46.34) 

digitalization 0.0171***

(57.80)

Industry Yes Yes Yes

Time Yes yes Yes

Control Yes Yes Yes

Constant -6.3738***

(-16.20) -6.8040***

(-18.03) -5.3799***

(-13.74)

Obs. 10883 10883 10883

R2 0.1969 0.1102 0.1766

Note: (1) ***, ** and * represent significant at 1%, 5% and 10% levels respectively. (2) The values in parentheses are Z value.

It can be found that in Table 7, the significances of all these three dimensions reach the 1% significanct level. Further, we conduct the seemingly unrelated estimation test (SUEST) to compare whether differences of coefficients exist. As SUEST can only compare whether the coefficients differed between two groups [41], we perform pairwise comparisons between the coefficients of coverage, usage and digitalization. Results in Table 8 indicate that the p values are all significant at 1% level while the coefficients of these three dimensions are 0.0223, 0.0116 and 0.0171 respectively. Therefore, it can be concluded that coverage breadth plays a much stronger role, followed by digitalization level. The breadth of digital financial coverage is more conducive to improving broaden the boundaries of financial services, which can benefit enterprises in remote areas. [42] also proved that the larger the coverage breadth is, the more customers can be covered, which is also consistent with [43].

Table 8 Results of SUEST test

Variable Usage Digitalization East SOE Second Third

Coverage 0.000*** 0.000*** 

Usage 0.000*** 

Non-east 0.000*** 

Private 0.001*** 

First 0.1193 0.2220

Second 0.1287

Note: (1) ***, ** and * represent significant at 1%, 5% and 10% levels respectively.

4. Fourth, when analyzing the heterogeneity of Equity nature, is there a large gap between the coefficients of state-owned and private companies, and can it be concluded that the development of local digital finance has a greater impact on private companies? These issues require further argument.

Response: Thanks for the comment. This part has been carefully revised as followed:

[48] suggest that the financial system in China is dominated by the banking system which is controlled by the government indeed. Although state-owned enterprises may suffer from financing problems to some extent, due to financial subsidies and policy supports they received from the governments, state-owned enterprises are much easier to get financing. Besides, private companies are mainly small and medium-sized compared with state-owned enterprises, which means they have fewer assets and collateral. Furthermore, private enterprises have lower management efficiency and quality of financial information. Thus, it is difficult for private enterprises to get funds from traditional financial markets. While the development of digital finance can attract more investors and provide more targeted financial products and services at the same time, the financial demands of private enterprises can be met to a larger extent. Therefore, this paper believes private enterprises are more likely to conduct M&As activities if financial constraints can be alleviated. Similarly, the value of 0 is given to state-owned enterprises, otherwise the variable is set as 1. Accordingly, we propose Hypothesis 4: Compared with state-owned enterprises, the positive effects of the development of digital finance on M&As activities are more prominent for private enterprises.

Column (3) and (4) in Table 9 presents the results of the impact of digital finance on M&As activities across companies with different equity nature. It shows that the significance of both state-owned and private enterprises is at the 1% level and the p value of SUEST test is 0.001, which is positive and significant at 1% level. The coefficients of state-owned and private enterprises are 0.0253 and 0.0287 individually, therefore, the development of digital finance put greater influences on private enterprises as their financial demands are much larger, thus supporting H4. The result is also consistent with [49-51].

5. Fifth, In the analysis of industry-level heterogeneity, the coefficient of the primary industry is larger. Why is it said that the impact on the secondary and tertiary industries is greater than that of the primary industry.

Response: Thanks for the comment. Changes have been made as followed:

5.4 Sector, digital finance and M&As

According to “Guidelines for the classification of listed companies in China” (2012 Version) [52], we divide samples into three categories based on industry attribute and do regression tests. Specifically, companies in category A are classified into primary sector mainly agricultural enterprises in China. Industrial manufacturing industry enterprises belonging to category B are considered as secondary sector. The remaining enterprises are regarded as third sector. Currently, agricultural enterprises are at mature stage and have lower capital demands, providing funds for banks and other financial institutions instead. On the contrary, industrial companies need plenty of funds to operation, achieving sustainable development and transformation. Also, most of the non-financial tertiary industries are private enterprises and companies’ size are relatively small, which makes them difficult to get funds from traditional financial institutions. Besides, primary industry has fewer M&As activities compared with secondary and third industries due to their operation strategies. Thus, we propose Hypothesis 5: The positive effects of the development of digital finance on M&As activities are less prominent on primary industry compared with secondary and third industry.

The regression results are presented in Column (5), (6) and (7) of Table 9. It can be found that the significances of primary, secondary and tertiary industry are all at 1% level with the coefficients of 0.0431, 0.0268 and 0.0281 respectively. However, SUEST test in Table 8 shows that the none of the p values are significant with the figures of 0.1193, 0.2220 and 0.1287 individually, which means there is no difference in the impact of digital finance on different sectors. Thus, Hypothesis 5 should be not accepted. The results differed with current research [53], one possible explanation is that, according to statistics, the number of M&As in primary industry is relatively small compared with the other two industries. After setting several sample selection criterions, the eligible samples size is only 101, which is not large enough to find the difference across sectors and may lead to variations in results. Thus, it also points out the way for future analysis.

Dear reviewer 3:

Thank you so much for your comments on the manuscript entitled “Digital finance and M&As: An empirical study and mechanism analysis” (Manuscript ID: PONE-D-23-09479). Your comments are very valuable and helpful for revising the paper and improving the quality. Details of revision are provided below.

1. The abstract of the paper is missing information about the method used.

Response: Thanks for the comment. We have extended the abstract. The revised part is presented below:

With the rapid growth and wide application of digital technology, enterprises have entered in the digital era with both opportunities and challenges existing. Mergers and acquisitions are one of the most efficient ways to integrate resources and achieve profit growth, gives enterprises advantages in competing in the new mode of economic growth. Based on this, this research tries to explore whether the development of digital finance will contribute to the emergence of M&As activities through combining M&As data of Chinese stock market with digital finance inclusion index between 2012 and 2020. The results show that the development of digital finance largely influence M&As activities through lower acquirers’ financial constraints. We further replace digital finance with three sub-indexes including coverage breadth, usage depth and digitalization level to explore the impact of different dimensions of digital finance on M&As. Results show that coverage breadth plays a more important role. In addition, heterogeneity tests reveal that the relationship between the development of digital finance and M&As activities varies significantly. The influence of digital finance on private and western and central enterprises are more significant compared with state-owned and eastern enterprises. According to the study, since the development of digital finance can be an efficient way to ease financial constraints and boost M&As activities, the government should promote the development of digital finance while companies strive to make the most use of it.

2. In-text citations are absolutely incorrect, they did not follow any standard. You can either use author+year or the number in the brackets - please revise throughout the paper. 

Response: Thanks for the comment. In-text citations have been thoroughly revised, please have a look.

3. Introduction - this section of the paper provides all necessary information about the previous papers, main aim and contribution (please, correct only the in-text citations).

Response: Thanks for the comment. In-text citations have been corrected.

4. Theory and hypotheses development - I do appreciate this section of the paper, which uses current references to develop the research hypotheses. 

Response: Thanks so much for your compliment.

5. Research and methods - please, clarify why 2012-2020 was chosen. Why is 2020 included? Is not this year significantly affected by the COVID-19 pandemic?

It would be appropriate to present some information about the dataset (basic descriptive characteristics).

Section 3.2 does not need to be too structured. 

Equations 3, 4, 5 - please explain all variables.

The methodological steps are missing, there is no information about the methods used in this section. Please add. 

Response:

Please, clarify why 2012-2020 was chosen. Why is 2020 included? Is not this year significantly affected by the COVID-19 pandemic?

After careful consideration, we would like to keep the data of 2020. There are several reasons, first, M&As activities are normally announced several months before the implementation time. Thus, the successful transaction in 2020 usually started before the COVID-19 pandemic, making the influence limited. Second, robustness test indicates that after delete the samples in 2020, the results does not change, which means the selected samples in 2020 are robust. A possible reason may be the most affected companies have failed to implement mergers and acquisitions activities.

It would be appropriate to present some information about the dataset (basic descriptive characteristics).

A summary of descriptive features has been added as followed:

The descriptive statistical results of the main variables in this research are shown in Table 2. It can be found that digitization grew rapidly during 2012 and 2020. The mean value of digital finance is 265.8168, with the maximum value being 431.9276, the minimum value being 18.84, and the standard deviation being 67.6133. There are significant differences in financial constraints among listed companies. The maximum value of financial constraints is -0.6839 while the minimum value is -5.1025, and the mean value is -3.7550. the statistical results of the other control variables, such as leverage and profitability, are close to the existing research results and are within a reasonable range.

Table 2 Summary of descriptive features

Variables Min Max Mean Std.

DF 18.84 431.9276 265.8168 67.6133

SA -5.1025 -0.6839 -3.7550 0.2692

Proportion 1 100 56.5613 34.8810

Market value 8.6399 13.3656 10.0004 0.5403

Age 4 49 17.7833 5.7756

Lev -1.6242 6270.051 3.2576 76.4856

EBIT 3.8942 11.5602 8.4541 0.6573

Ownership 8.26 100 59.8881 14.8458

Separation 0 1 0.3112 0.4630

Section 3.2 does not need to be too structured. 

Section 3.2 has been restructured, please have a look.

Equations 3, 4, 5 - please explain all variables.

Changes have been made as followed:

 (1)

Where Size is the natural logarithm of the company’s total assets, while Age is the difference between the observation year and the registration year of the company. Therefore, the outcome of the SA index is less than 0, the larger the score is, the greater the financing constraints will be.

Probit(〖MA〗_it=1)=α_1+α_2 DF_it+∑▒〖α_3 Controls_it+∑▒〖year+∑▒ind〗〗+ε_1 (2)

SA=β_1+β_2 〖DF〗_it+∑▒〖β_3 Controls_it+∑▒〖year+∑▒ind〗〗+ε_2 (3)

Probit(〖MA〗_it=1)=θ_1+θ_2 DF_it+∑▒〖θ_3 Controls_it+∑▒〖year+∑▒ind〗〗+ε_3 (4)

In the above equations, subscripts i and t represent industry and time respectively. The dependent variable MA is. The independent variable DF represents the level of digital finance development in company’s province in year t. While controls refer to the control variables, and ε is the random error. In addition, this paper also fixes the effects of industry and time effects to control for possible interference in the regression results of unobserved factors at different levels. The coefficient α_2, β_2, θ_2 of the independent variable DF_it reflects the overall impact of the development of digital finance on M&As activities, which is expected to be significantly positive according to the hypothesis above.

The methodological steps are missing, there is no information about the methods used in this section. Please add. 

The methodology used in this research has been added as followed:

Before running linear regression test, we first adopt VIF correlation test to examine whether there are multicollinearity issues. In order to examine whether the development of digital finance will contribute to M&A activities and given that the dependent variable is a dummy variable, this research has selected the binary Probit model as the base model to test H1, and the regression model (2) is set below. To examine H2, this research uses bootstrap method to examine the mediation effect and the regression model is presented in equation (2) to (4). If 0 is not contained in the confidence interval, then it can be concluded that the mediating effect of financial constraints exists.

6. Were the preconditions of the regression analysis verified? Please give details. 

Response: Correlation test has been run to verify the preconditions of the regression analysis and listed below.

Generally speaking, there may be multicollinearity issue when the vif between explanatory variables is higher than 10. The results in Table 3 indicate that all the value of vif is below 10, which can prove that there is no multicollinearity issue existing in this study and the regression model would not be affected.

Table 3 VIF correlation test

Variable VIF 1/VIF

MV 4.33 0.231029

EBIT 3.99 0.250522

Age 3.23 0.309354

SA 3.14 0.318314

Ownership 1.12 0.891274

Separation 1.04 0.962070

Proportion 1.01 0.991414

Lev 1.00 0.997976

Mean VIF 2.36

Note: ** and * represent significance at the 1% and 5% levels respectively.

7. The results are explained very succinctly - please check and revise.

Response: Regression results have been explained more detailed as followed:

 Regression analysis

The influence of the development of digital finance on merger and acquisition activities are shown in Table 4 below. Model (1) only considers the relationship between digital finance and M&As without adding other control variables while Model (2) is the result with control variables. It can be found that the coefficient of DF is 0.0044 and 0.0050 respectively and significantly positive at 1% significance level, indicating that the development of digital finance can largely contribute to the occurrence of M&As activities. Therefore, hypothesis 1 is true.

4.4 Mediating analysis

The results of mediating analysis are represented in Table 5 below. It can be found that the coefficients of DF in both model (2) and (3) are significant at 1% level. After adding financial constraints in the model (4), the result is still significant at 1% level. In addition, according to bootstrap examination results, the confidence interval of indirect effect is between 0.0005 and 0.007 and 0 is not included, which indicates that financial constraints play a mediating effect in how the development of digital finance contributes to M&As activities. Thus, H2 has been verified.

8. Please, focus more on the discussion section, use more references to discuss your findings in the context of other relevant studies published online.

Response: discussion section has been fully revised as followed:

Considering the heterogeneity of digital financial structure, the imbalance of regional economic development, differences in enterprises’ equity nature and industries, these may lead to great differences in the impact of the digital finance on M&As activities in different enterprises. Therefore, this research divides digital finance into three dimensions and classifies the samples into several categories according to regions , equity nature and sector differences to do further analysis.

 The heterogeneity of digital financial structure

Digital finance index covers three sub-dimensions including coverage breadth, usage depth and digitalization level. Coverage breadth reflects the regional financial environment by counting the number of people using electronic accounts. Usage depth describes the regional financial services ability through the usage of various digital financial services while digitalization level measures the cost and efficiency of digital finance. In order to more accurately examine which dimension plays a more significant role on enterprises’ M&As activities, we carry out structural analysis of digital finance. The results are shown in Table 7.

Table 7 Results of the heterogeneity of digital financial structure

Variable Model (10) Model (11) Model (12)

coverage 0.0223***

(60.84) 

usage 0.0116***

(46.34) 

digitalization 0.0171***

(57.80)

Industry Yes Yes Yes

Time Yes yes Yes

Control Yes Yes Yes

Constant -6.3738***

(-16.20) -6.8040***

(-18.03) -5.3799***

(-13.74)

Obs. 10883 10883 10883

R2 0.1969 0.1102 0.1766

Note: (1) ***, ** and * represent significant at 1%, 5% and 10% levels respectively. (2) The values in parentheses are Z value.

It can be found that in Table 7, the significances of all these three dimensions reach the 1% significanct level. Further, we conduct the seemingly unrelated estimation test (SUEST) to compare whether differences of coefficients exist. As SUEST can only compare whether the coefficients differed between two groups [41], we perform pairwise comparisons between the coefficients of coverage, usage and digitalization. Results in Table 8 indicate that the p values are all significant at 1% level while the coefficients of these three dimensions are 0.0223, 0.0116 and 0.0171 respectively. Therefore, it can be concluded that coverage breadth plays a much stronger role, followed by digitalization level. The breadth of digital financial coverage is more conducive to improving broaden the boundaries of financial services, which can benefit enterprises in remote areas. [42] also proved that the larger the coverage breadth is, the more customers can be covered, which is also consistent with [43].

Table 8 Results of SUEST test

Variable Usage Digitalization East SOE Second Third

Coverage 0.000*** 0.000*** 

Usage 0.000*** 

Non-east 0.000*** 

Private 0.001*** 

First 0.1193 0.2220

Second 0.1287

Note: (1) ***, ** and * represent significant at 1%, 5% and 10% levels respectively.

 Region, digital finance and M&As

The unbalanced distribution of financial resources broadens the economic gap between different regions, which leads to the east more developed while the central and west less developing. Besides, according to “China regional financial operation report (2021)” [44] issued by The People's Bank of China, the number of banking financial institutions in western and middle China was 59,000 and 53,000 respectively, compared with more 100,000 in the east. As a result, the financial demand of most enterprises especially in west and central area cannot be met.

According to the criteria for the division of east, west and central areas in China issued by National Bureau of Statistics, we divide samples into two categories based on the regions where companies registered. 0 is given to companies located in eastern region, while central and western areas are set as 1. Based on the analysis above, this paper proposes Hypothesis 3: The influences of digital finance on M&As activities are more significant for enterprises located in west and central areas.

As the Column (1) - (2) shown in Table 9, it can be found that the significances of both east and non-east areas are at 1% significant level. SUEST test indicates that the p value is 0.000 and significant at 1% level, which proves the influences various on different areas. Since the coefficients of east and Midwest areas are 0.0379 and 0.1008 separately, it can be concluded that the effects of the development of digital finance in promoting M&As activities is much stronger in the west and central area. Thus, H3 is approved. After receiving same level of financial supports, Midwest enterprises are more likely to implement M&As activities, which is also consistent with [20], [45-47].

 Table 9 Regional heterogeneity test results

Variable Region Equity nature Sector

 (1)

East (2)

Non-east (3)

SOE (4)

Private (5)

First (6)

Second (7)

Third

DF 0.0379***

(62.23) 0.1008***

(38.21) 0.0253***

(34.05) 0.0287***

(55.87) 0.0431***

(6.38) 0.0268***

(52.99) 0.0281***

(37.53)

Control Yes Yes Yes Yes Yes Yes Yes

Industry Yes Yes Yes Yes Yes Yes Yes

Time Yes Yes Yes Yes Yes Yes Yes

Constant -8.3418***

(-15.36) -9.7294***

(-6.50) -4.7314***

(-6.07) -7.3064***

(-12.85) -1.7503

(-0.29) -9.3662***

(-8.60) -5.1855***

(-5.76)

Obs. 8098 2785 3217 7666 101 7101 3681

R2 0.3105 0.5964 0.2120 0.2494 0.3032 0.2322 0.2436

Note: (1) ***, ** and * represent significant at 1%, 5% and 10% levels respectively. (2) The values in parentheses are Z value.

5.3 Equity nature, digital finance and M&As

[48] suggest that the financial system in China is dominated by the banking system which is controlled by the government indeed. Although state-owned enterprises may suffer from financing problems to some extent, due to financial subsidies and policy supports they received from the governments, state-owned enterprises are much easier to get financing. Besides, private companies are mainly small and medium-sized compared with state-owned enterprises, which means they have fewer assets and collateral. Furthermore, private enterprises have lower management efficiency and quality of financial information. Thus, it is difficult for private enterprises to get funds from traditional financial markets. While the development of digital finance can attract more investors and provide more targeted financial products and services at the same time, the financial demands of private enterprises can be met to a larger extent. Therefore, this paper believes private enterprises are more likely to conduct M&As activities if financial constraints can be alleviated. Similarly, the value of 0 is given to state-owned enterprises, otherwise the variable is set as 1. Accordingly, we propose Hypothesis 4: Compared with state-owned enterprises, the positive effects of the development of digital finance on M&As activities are more prominent for private enterprises.

Column (3) and (4) in Table 9 presents the results of the impact of digital finance on M&As activities across companies with different equity nature. It shows that the significance of both state-owned and private enterprises is at the 1% level and the p value of SUEST test is 0.001, which is positive and significant at 1% level. The coefficients of state-owned and private enterprises are 0.0253 and 0.0287 individually, therefore, the development of digital finance put greater influences on private enterprises as their financial demands are much larger, thus supporting H4. The result is also consistent with [49-51].

5.4 Sector, digital finance and M&As

According to “Guidelines for the classification of listed companies in China” (2012 Version) [52], we divide samples into three categories based on industry attribute and do regression tests. Specifically, companies in category A are classified into primary sector mainly agricultural enterprises in China. Industrial manufacturing industry enterprises belonging to category B are considered as secondary sector. The remaining enterprises are regarded as third sector. Currently, agricultural enterprises are at mature stage and have lower capital demands, providing funds for banks and other financial institutions instead. On the contrary, industrial companies need plenty of funds to operation, achieving sustainable development and transformation. Also, most of the non-financial tertiary industries are private enterprises and companies’ size are relatively small, which makes them difficult to get funds from traditional financial institutions. Besides, primary industry has fewer M&As activities compared with secondary and third industries due to their operation strategies. Thus, we propose Hypothesis 5: The positive effects of the development of digital finance on M&As activities are less prominent on primary industry compared with secondary and third industry.

The regression results are presented in Column (5), (6) and (7) of Table 9. It can be found that the significances of primary, secondary and tertiary industry are all at 1% level with the coefficients of 0.0431, 0.0268 and 0.0281 respectively. However, SUEST test in Table 8 shows that the none of the p values are significant with the figures of 0.1193, 0.2220 and 0.1287 individually, which means there is no difference in the impact of digital finance on different sectors. Thus, Hypothesis 5 should be not accepted. The results differed with current research [53], one possible explanation is that, according to statistics, the number of M&As in primary industry is relatively small compared with the other two industries. After setting several sample selection criterions, the eligible samples size is only 101, which is not large enough to find the difference across sectors and may lead to variations in results. Thus, it also points out the way for future analysis.

---

## [Decision Letter · Decision Letter 1]

29 Jun 2023

PONE-D-23-09479R1Digital finance and M&As: An empirical study and mechanism analysisPLOS ONE

Dear Dr. ma,

Thank you for submitting your manuscript to PLOS ONE. After careful consideration, we feel that it has merit but does not fully meet PLOS ONE’s publication criteria as it currently stands. Therefore, we invite you to submit a revised version of the manuscript that addresses the points raised during the review process.

We look forward to receiving your revised manuscript.

Kind regards,

Stefan Cristian Gherghina, PhD. Habil.

Academic Editor

PLOS ONE

Journal Requirements:

Additional Editor Comments (if provided):

The remaining comments of the second referee should be further addressed.

Reviewers' comments:

Reviewer's Responses to Questions

**Comments to the Author**

1. If the authors have adequately addressed your comments raised in a previous round of review and you feel that this manuscript is now acceptable for publication, you may indicate that here to bypass the “Comments to the Author” section, enter your conflict of interest statement in the “Confidential to Editor” section, and submit your "Accept" recommendation.

Reviewer #1: All comments have been addressed

Reviewer #3: (No Response)

Reviewer #4: (No Response)

2. Is the manuscript technically sound, and do the data support the conclusions?

Reviewer #1: Yes

Reviewer #3: Yes

Reviewer #4: Yes

3. Has the statistical analysis been performed appropriately and rigorously? 

Reviewer #1: Yes

Reviewer #3: Yes

Reviewer #4: Yes

4. Have the authors made all data underlying the findings in their manuscript fully available?

Reviewer #1: Yes

Reviewer #3: Yes

Reviewer #4: Yes

5. Is the manuscript presented in an intelligible fashion and written in standard English?

Reviewer #1: Yes

Reviewer #3: Yes

Reviewer #4: Yes

6. Review Comments to the Author

Reviewer #1: The topic is meaningful to prove how the digital finance affects M&As, the authors had revised this paper by each comment seriously.

Reviewer #3: Dear authors, Thank you very much for the revisions. I only have two additional comments:

1. Revise the paper again to reduce the grammar and punctuation imperfections;

2. It would be beneficial to extend the discussion section and compare your outputs and findings in a broader context.

Reviewer #4: (No Response)

7. PLOS authors have the option to publish the peer review history of their article (what does this mean?). If published, this will include your full peer review and any attached files.

Reviewer #1: No

Reviewer #3: No

Reviewer #4: No

---

## [Author Response · Author response to Decision Letter 1]

15 Jul 2023

Rebuttal letter

Dear editor:

Thank you so much for your advice on the manuscript entitled “Digital finance and M&As: An empirical study and mechanism analysis” (Manuscript ID: PONE-D-23-09479). Your comments are very valuable and helpful for revising the paper and improving the quality. As you mentioned, this paper referred retracted articles. We have substituted these two references with other articles, please have a check. Let me know if there are further questions.

Dear reviewer 1:

Thank you so much for your comments about the abstract, which is quite beneficial for improving the paper quality. We are happy to know to that our revised paper meets the criteria.

Dear reviewer 3:

Thank you so much for your comments on the manuscript entitled “Digital finance and M&As: An empirical study and mechanism analysis” (Manuscript ID: PONE-D-23-09479). Your comments are very valuable and helpful for revising the paper and improving the quality. Details of revision are provided below.

1. Revise the paper again to reduce the grammar and punctuation imperfections;

Response: Thanks for the comments, we have revised the paper thoroughly again, please have a look.

2. It would be beneficial to extend the discussion section and compare your outputs and findings in a broader context.

Response: Thanks for the comments, the discussion has been extended as followed:

5. Further discussion

Considering the heterogeneity of digital financial structure, the imbalance of regional economic development, and differences in enterprises’ equity nature and industries, these may lead to great differences in the impact of digital finance on M&As activities in different enterprises. Therefore, this research divides digital finance into three dimensions and classifies the samples into several categories according to regions, equity nature, and sector differences to do further analysis.

5.1 The heterogeneity of digital financial structure

The digital finance index covers three sub-dimensions including coverage breadth, usage depth, and digitalization level. Coverage breadth reflects the regional financial environment by counting the number of people using electronic accounts. Usage depth describes the regional financial services’ ability through the usage of various digital financial services while digitalization level measures the cost and efficiency of digital finance. In order to more accurately examine which dimension plays a more significant role on enterprises’ M&As activities, we carry out structural analysis of digital finance. The results are shown in Table 7.

Table 7 Results of the heterogeneity of digital financial structure

Variable Model (10) Model (11) Model (12)

coverage 0.0223***

(60.84) 

usage 0.0116***

(46.34) 

digitalization 0.0171***

(57.80)

Industry Yes Yes Yes

Time Yes yes Yes

Control Yes Yes Yes

Constant -6.3738***

(-16.20) -6.8040***

(-18.03) -5.3799***

(-13.74)

Obs. 10883 10883 10883

R2 0.1969 0.1102 0.1766

Note: (1) ***, **, and * represent significant at 1%, 5%, and 10% levels respectively. (2) The values in parentheses are Z values.

It can be found that in Table 7, the significances of all these three dimensions reach the 1% significant level. Further, we conduct the seemingly unrelated estimation test (SUEST) to compare whether differences in coefficients exist. As SUEST can only compare whether the coefficients differed between two groups [41], we perform pairwise comparisons between the coefficients of coverage, usage, and digitalization. Results in Table 8 indicate that the p values are all significant at 1% level while the coefficients of these three dimensions are 0.0223, 0.0116, and 0.0171 respectively. Therefore, it can be concluded that coverage breadth plays a much stronger role, followed by digitalization level. The breadth of digital financial coverage is more conducive to improving broaden the boundaries of financial services and from the perspective of efficiency, coverage breadth is the most direct way to benefit enterprises in remote areas. Therefore, more digital infrastructure should be constructed at the early stage, through expanding the coverage of digital services to strengthen usage depth and promote digitalization level. [42] also proved that the larger the coverage breadth is, the more customers can be covered, which is also proved by [43].

Table 8 Results of SUEST test

Variable Usage Digitalization East SOE Second Third

Coverage 0.000*** 0.000*** 

Usage 0.000*** 

Non-east 0.000*** 

Private 0.001*** 

First 0.1193 0.2220

Second 0.1287

Note: (1) ***, **, and * represent significant at 1%, 5%, and 10% levels respectively.

5.2 Region, digital finance and M&As

The unbalanced distribution of financial resources broadens the economic gap between different regions, which leads to the east more developed while the central and west being less developing. Besides, according to “China regional financial operation report (2021)” [44] issued by The People's Bank of China, the number of banking financial institutions in western and middle China was 59,000 and 53,000 respectively, compared with more than 100,000 in the east. As a result, the financial demand of most enterprises especially in west and central areas cannot be met.

According to the criteria for the division of east, west, and central areas in China issued by the National Bureau of Statistics, we divide samples into two categories based on the regions where companies registered. 0 is given to companies located in eastern region, while central and western areas are set as 1. Based on the analysis above, this paper proposes Hypothesis 3: The influences of digital finance on M&As activities are more significant for enterprises located in west and central areas.

As Column (1) - (2) shown in Table 9, it can be found that the significances of both east and non-east areas are at 1% significant level. SUEST test indicates that the p value is 0.000 and significant at 1% level, which proves the influences various on different areas. Since the coefficients of east and Midwest areas are 0.0379 and 0.1008 separately, it can be concluded that the effects of the development of digital finance in promoting M&As activities are much stronger in the west and central area. Thus, H3 is approved. After receiving the same level of financial supports, Midwest enterprises are more likely to implement M&As activities. Therefore, the promotion of the development of digital finance is important for narrowing the regional financial gap and give Midwest companies more opportunities for achieving sustainable development and industrial transformation. The findings are also consistent with [20], [45-49].

Table 9 Regional heterogeneity test results

Variable Region Equity nature Sector

 (1)

East (2)

Non-east (3)

SOE (4)

Private (5)

First (6)

Second (7)

Third

DF 0.0379***

(62.23) 0.1008***

(38.21) 0.0253***

(34.05) 0.0287***

(55.87) 0.0431***

(6.38) 0.0268***

(52.99) 0.0281***

(37.53)

Control Yes Yes Yes Yes Yes Yes Yes

Industry Yes Yes Yes Yes Yes Yes Yes

Time Yes Yes Yes Yes Yes Yes Yes

Constant -8.3418***

(-15.36) -9.7294***

(-6.50) -4.7314***

(-6.07) -7.3064***

(-12.85) -1.7503

(-0.29) -9.3662***

(-8.60) -5.1855***

(-5.76)

Obs. 8098 2785 3217 7666 101 7101 3681

R2 0.3105 0.5964 0.2120 0.2494 0.3032 0.2322 0.2436

Note: (1) ***, **, and * represent significant at 1%, 5%, and 10% levels respectively. (2) The values in parentheses are Z values.

5.3 Equity nature, digital finance, and M&As

[50] suggest that the financial system in China is dominated by the banking system which is controlled by the government indeed. Although state-owned enterprises may suffer from financing problems to some extent, due to financial subsidies and policy supports they received from the governments, state-owned enterprises are much easier to get financing. Besides, private companies are mainly small and medium-sized compared with state-owned enterprises, which means they have fewer assets and collateral. Furthermore, private enterprises have lower management efficiency and quality of financial information. Thus, it is difficult for private enterprises to get funds from traditional financial markets. While the development of digital finance can attract more investors and provide more targeted financial products and services at the same time, the financial demands of private enterprises can be met to a larger extent. Therefore, this paper believes private enterprises are more likely to conduct M&As activities if financial constraints can be alleviated. Similarly, the value of 0 is given to state-owned enterprises, otherwise, the variable is set as 1. Accordingly, we propose Hypothesis 4: Compared with state-owned enterprises, the positive effects of the development of digital finance on M&As activities are more prominent for private enterprises.

Column (3) and (4) in Table 9 presents the results of the impact of digital finance on M&As activities across companies with different equity nature. It shows that the significance of both state-owned and private enterprises is at the 1% level and the p value of SUEST test is 0.001, which is positive and significant at 1% level. The coefficients of state-owned and private enterprises are 0.0253 and 0.0287 individually, therefore, the development of digital finance put greater influence on private enterprises as their financial demands are much larger, thus supporting H4. Private enterprises make up an important role of China’s economy and the digital finance enhance their access to financial services. As a result, private enterprises operate and produce more efficiently, which leads to a more dynamic economy. The result is also consistent with [51-54].

5.4 Sector, digital finance, and M&As

According to “Guidelines for the classification of listed companies in China” (2012 Version) [55], we divide samples into three categories based on industry attributes and do regression tests. Specifically, companies in category A are classified into primary sector mainly agricultural enterprises in China. Industrial manufacturing industry enterprises belonging to category B are considered as secondary sector. The remaining enterprises are regarded as third sector. Currently, agricultural enterprises are at the mature stage and have lower capital demands, providing funds for banks and other financial institutions instead. On the contrary, industrial companies need plenty of funds to operation to achieve sustainable development and industrial transformation. Also, most of the non-financial tertiary industries are private enterprises and companies’ sizes are relatively small, which makes them difficult to get funds from traditional financial institutions. Besides, primary industry has fewer M&As activities compared with secondary and third industries due to their operation strategies. Thus, we propose Hypothesis 5: The positive effects of the development of digital finance on M&As activities are less prominent on primary industry compared with secondary and third industry.

The regression results are presented in Column (5), (6), and (7) of Table 9. It can be found that the significances of primary, secondary, and tertiary industry are all at 1% level with the coefficients of 0.0431, 0.0268, and 0.0281 respectively. However, SUEST test in Table 8 shows that none of the p values are significant with the figures of 0.1193, 0.2220, and 0.1287 individually, which means there is no difference in the impact of digital finance on different sectors. Thus, Hypothesis 5 should be not accepted. One possible explanation is that, according to statistics, the number of M&As in the primary industry is relatively small compared with the other two industries. After setting several sample selection criterions, the eligible sample size is only 101, which is not large enough to find the difference across sectors and may lead to variations in results. While a recent research [56] indicates that the impact of the development of digital finance on M&As activities varies across industries. To be more specific, the positive effects of the development of digital finance on M&As activities are more prominent in the tertiary industry with the primary coming to the second. However, their findings still differ from our hypothesis. Thus, it points out the way for future analysis.

---

## [Decision Letter · Decision Letter 2]

27 Jul 2023

Digital finance and M&As: An empirical study and mechanism analysis

PONE-D-23-09479R2

Dear Dr. ma,

We’re pleased to inform you that your manuscript has been judged scientifically suitable for publication and will be formally accepted for publication once it meets all outstanding technical requirements.

Kind regards,

Stefan Cristian Gherghina, PhD. Habil.

Academic Editor

PLOS ONE

Additional Editor Comments (optional):

Reviewers' comments:

Reviewer's Responses to Questions

**Comments to the Author**

1. If the authors have adequately addressed your comments raised in a previous round of review and you feel that this manuscript is now acceptable for publication, you may indicate that here to bypass the “Comments to the Author” section, enter your conflict of interest statement in the “Confidential to Editor” section, and submit your "Accept" recommendation.

Reviewer #1: All comments have been addressed

Reviewer #3: All comments have been addressed

Reviewer #4: All comments have been addressed

2. Is the manuscript technically sound, and do the data support the conclusions?

Reviewer #1: Yes

Reviewer #3: Yes

Reviewer #4: Yes

3. Has the statistical analysis been performed appropriately and rigorously? 

Reviewer #1: Yes

Reviewer #3: Yes

Reviewer #4: Yes

4. Have the authors made all data underlying the findings in their manuscript fully available?

Reviewer #1: Yes

Reviewer #3: Yes

Reviewer #4: Yes

5. Is the manuscript presented in an intelligible fashion and written in standard English?

Reviewer #1: Yes

Reviewer #3: Yes

Reviewer #4: Yes

6. Review Comments to the Author

Reviewer #1: I feel that this manuscript is now acceptable for publication since all comments had been addressed carefully.

Reviewer #3: (No Response)

Reviewer #4: The research topic of this manuscript is very meaningful, with a reasonable research process and clear conclusions. I think this manuscript can be accepted by the journal.

7. PLOS authors have the option to publish the peer review history of their article (what does this mean?). If published, this will include your full peer review and any attached files.

Reviewer #1: No

Reviewer #3: No

Reviewer #4: No

---

## [Editor Report · Acceptance letter]

31 Jul 2023

PONE-D-23-09479R2 

Digital finance and M&As: An empirical study and mechanism analysis 

Dear Dr. Ma:

I'm pleased to inform you that your manuscript has been deemed suitable for publication in PLOS ONE. Congratulations! Your manuscript is now with our production department. 

Kind regards, 

on behalf of

Dr. Stefan Cristian Gherghina 

Academic Editor

PLOS ONE